# Quantum Mechanical Approach to the Khintchine and Bochner Criteria for Characteristic Functions

**DOI:** 10.3390/e25071042

**Published:** 2023-07-11

**Authors:** Leon Cohen

**Affiliations:** Department of Physics, Hunter College of the City University of New York, 695 Park Ave., New York, NY 10065, USA; leon.cohen@hunter.cuny.edu

**Keywords:** quantum probability, characteristic function, Khintchine criteria, Bochner’s theorem

## Abstract

While it is generally accepted that quantum mechanics is a probability theory, its methods differ radically from standard probability theory. We use the methods of quantum mechanics to understand some fundamental aspects of standard probability theory. We show that wave functions and operators do appear in standard probability theory. We do so by generalizing the Khintchine and Bochner criteria for a complex function to be a characteristic function. We show that quantum mechanics clarifies these criteria and suggests generalizations of them.

## 1. Introduction

It is remarkable that standard probability theory, developed over the last 300 years and with immense successful applications in almost all areas of science and engineering is dramatically different from the most successful probability theory, namely quantum mechanics. How is that possible? There are no wave functions or operators in classical probability theory, while they are fundamental in quantum mechanics. We explore the possibility that the two theories have commonalities, and in particular, we show that wave functions and operators do, in fact, appear in standard probability theory. We do so by rewriting the Khintchine and Bochner criteria of standard probability theory, which are necessary and sufficient conditions for a complex function to be a proper characteristic function. In addition, we show that quantum mechanics clarify these criteria and suggests generalizations of them.

## 2. Quantum Mechanical Random Variables and Probability Densities

For notational clarity, we discuss the fundamental issues of quantum mechanics central to our subsequent discussion of characteristic functions. In quantum mechanics, one associates operators with observables [1,2]. The numerical values for the observable are obtained by solving the eigenvalue problem for the operator (Operators are denoted in boldface, and all integrals go from −∞ to ∞ unless otherwise noted), A,
(1)$$Au_{a}\left(x\right)=au_{a}\left(x\right)$$
where *a* are the eigenvalues and $u_{a}\left(x\right)$ are their corresponding eigenfunctions. In writing Equation (Equation 1) we assume that the eigenvalues are continuous. The discrete case may be straightforwardly obtained from the continuous case. From the usual probability point of view, there are three fundamental idea relevant to our considerations.

**Random variables.** In quantum mechanics, the random variables are the eigenvalues. They have to be real, and that is assured if the operator A is Hermitian. Additionally, if the operator is Hermitian, the eigenfunctions are complete and orthogonal
(2)$$\begin{array}{cc} \int \;u_{a^{\prime }}^{*}\left(x\right)u_{a}\left(x\right)\;dx\; & =\;\delta (a-a^{\prime }) \end{array}$$
(3)$$\begin{array}{cc} \int \;\;u_{a}^{*}\left(x^{\prime }\right)u_{a}\left(x\right)\;da\; & =\;\delta (x-x^{\prime }) \end{array}$$

**Quantum probability densities**. To obtain the probability density corresponding to the random variables, one expands the position wave function, ψ(x), as
(4)$$\psi \left(x\right)=\int c\left(a\right)u_{a}\left(x\right)\;da$$
where the “expansion function”, c(a), is given by
(5)$$c\left(a\right)=\int \psi \left(x\right)u_{a}^{*}\left(x\right)\;dx$$
The expansion function, c(a), is the wave function in the *a* representation and the probability density for the random variable *a* is then
(6)$$P\left(a\right)={\left|c(a)\right|}^{2}$$
This is a crucial aspect of quantum mechanics devised by Born in 1926. It is radically different from the standard method of transforming probability density functions.

**Expectation values.** may be calculated in two different ways. Since ${\left|c(a)\right|}^{2}$ is the probability density, by the usual definitions of expectation value we have
(7)$$\langle a\rangle =\int a{\left|c(a)\right|}^{2}da$$
However, one can also calculate it by way of
(8)$$\langle A\rangle =\int \psi^{*}\left(x\right)A\psi \left(x\right)dx$$
That Equations (Equation 7) and (Equation 8) are equivalent
(9)〈a〉=〈A〉
is easily proven by inserting Equation (Equation 4) into Equation (Equation 8).

## 3. Standard Characteristic Function

In standard probability theory, the characteristic function for the random variable *x* corresponding to a probability distribution P(x) is defined [3,4]
(10)$$M_{x}\left(\theta \right)=\int e^{i\theta x}P\left(x\right)\;dx$$
From the characteristic function, one may obtain the probability density by Fourier inversion,
(11)$$P\left(x\right)=\frac{1}{2\pi }\int \;M\left(\theta \right)e^{-i\theta x}d\theta$$
The characteristic function is the expectation value of $e^{i\theta x}.$
(12)$$M_{x}\left(\theta \right)=\langle e^{i\theta x}\rangle$$
A fundamental aspect of characteristic functions is that expectation values may be obtained by differentiation. In particular, the moments are given by
(13)$$\langle x^{n}\rangle =\frac{1}{i^{n}}\frac{d^{n}}{d\theta^{n}}M_{x}\left(\theta \right)\;{|}_{\theta =0}$$
If the moments are known, the characteristic function can be constructed by way of
(14)$$M_{x}\left(\theta \right)=\sum_{n=0}^{\infty }\frac{{\left(i\theta \right)}^{n}}{n!}\langle x^{n}\rangle$$
However, we point out that the moments do not always determine a probability density function uniquely. Such densities are said to be “moment-indeterminate”, or “M-indeterminate” and quantum mechanics has elucidated some issues in that regard [5,6,7,8].

### Necessary and Sufficient Conditions for a Function to Be a
Characteristic Function

A historically important question in probability theory has been finding necessary and sufficient conditions for a complex function to be a characteristic function. The two best known criteria are that of Khintchine and Bochner that we discuss in the following sections [3,4,9,10]. We mention here that there are some obvious conditions that a characteristic satisfies, namely M(0)=1 and $M^{*}\left(-\theta \right)=M\left(\theta \right).$

## 4. Khintchine Criteria, Quantum Mechanics, and Its Generalization

The Khintchine criteria is that a complex function, M(θ), is a characteristic function if and only if it admits the representation
(15)$$M_{x}\left(\theta \right)=\int g^{*}\left(x\right)g\left(x+\theta \right)dx$$
for some functions, g(x), which is the normalized one
(16)$$\int {\left|g(x)\right|}^{2}dx=1$$

While the Khintchine theorem is fundamental in probability theory, the nature of the g(x) functions are seldom discussed, and the question of uniqueness is rarely mentioned. We now discuss Equation (Equation 15) from a quantum mechanical point of view; this will allow us to generalize the criteria and show that the *g*’s behave akin to wave functions, are not unique and an infinite number are readily generated.

To understand the criteria from a quantum mechanical viewpoint, we calculate the probability density by way of Equation (Equation 11). In anticipation of the results, we use *p* for the random variable
(17)$$P\left(p\right)=\int g^{*}\left(x\right)g\left(x+\theta \right)e^{-i\theta p}dxd\theta$$
making the change of variables $x^{\prime }=x+\theta$; $dx^{\prime }=d\theta$ we obtain
(18)$$P\left(p\right)=\frac{1}{2\pi }\;\int \int \;{g^{*}\left(x\right)g(x}^{\prime })e^{-i(x^{\prime }-x)p}dxdx^{\prime }$$
which evaluates to
(19)$$P\left(p\right)={\left|\frac{1}{\sqrt{2\pi }}\int g\left(x\right)e^{-ixp}\right|}^{2}$$
If we write this as
(20)$$P\left(p\right)={\left|\varphi (p)\right|}^{2}$$
with
(21)$$\varphi \left(p\right)=\frac{1}{\sqrt{2\pi }}\int g\left(x\right)e^{-ixp}$$
then clearly φ(p) is the momentum wave function corresponding to the position wave function g(x). Equation (Equation 20) is the probability distribution of momentum. Why momentum? How did quantum mechanical momentum occur? We now rewrite the Khintchine theorem in terms of operators. Using the fact that
(22)$$g\left(x+\theta \right)=e^{\theta \frac{d}{dx}}g\left(x\right)$$
which is the case since $e^{\theta \frac{d}{dx}}$ is the translation operator [11]. We write it as
(23)$$e^{\theta \frac{d}{dx}}=e^{i\theta p}$$
where
(24)$$p=\frac{1}{i}\frac{d}{dx}$$
is the quantum mechanical momentum operator. Therefore we may write the Khintchine criteria Equation (Equation 15) as
(25)$$M\left(\theta \right)=\int g^{*}\left(x\right)e^{i\theta p}g\left(x\right)dx$$
In Equation (Equation 25) we see that M(θ) that is an expectation value, the expectation value of the operator $e^{i\theta p}$
(26)$$M\left(\theta \right)=\langle e^{i\theta p}\rangle$$
This makes sense since, indeed, in standard probability theory the characteristic function is given by Equation (Equation 12). However, we are calculating it from a quantum mechanics point of view as per Equation (Equation 8). In anticipation of our generalization, we define the characteristic function operator for momentum by
(27)$$M_{p}\left(\theta \right)=e^{i\theta p}$$
in which case the characteristic function is
(28)$$M_{p}\left(\theta \right)=\langle M_{p}\left(\theta \right)\rangle$$

### 4.1. Non-Uniqueness of g(x)

We now show that the function g(x) appearing in Equation (Equation 15) is not unique. The quantum viewpoint makes this clear. Let us suppose that $g_{1}\left(x\right)$ satisfies the Khintchine criteria and therefore, the associated characteristic function is
(29)$$M_{1}\left(\theta \right)=\int {g_{1}^{*}\left(x\right)g}_{1}(x+\theta )dx$$
The corresponding momentum wave, as given by Equation (Equation 21), is then
(30)$$\varphi_{1}\left(p\right)=\frac{1}{\sqrt{2\pi }}\int g_{1}\left(x\right)e^{-ixp}$$
Now, from a quantum mechanical point of view, we know the probability distribution is the absolute value squared of $\varphi_{1}\left(a\right)$ and therefore defining
(31)$$\varphi_{2}\left(p\right)=\varphi_{1}\left(p\right)e^{iS(p)}$$
where S(p) is an arbitrary real function gives the same probability distribution
(32)$${\left|\varphi_{2}\left(p\right)\right|}^{2}={\left|\varphi_{1}\left(p\right)\right|}^{2}$$
We now find the corresponding $g_{2}(x).$ We have
(33)$$\begin{array}{cc} \varphi_{2}\left(p\right) & =\varphi_{1}\left(p\right)e^{iS(p)} \end{array}$$
(34)$$\begin{array}{cc} \; & =\frac{1}{\sqrt{2\pi }}e^{iS(p)}\int g_{1}\left(x\right)e^{-ixp}dx \end{array}$$
(35)$$\begin{array}{cc} \; & =\frac{1}{\sqrt{2\pi }}\int g_{2}\left(x\right)e^{-ixp}dx \end{array}$$
Solving for $g_{2}\left(x\right)$ we obtain
(36)$$g_{2}\left(x\right)=\frac{1}{2\pi }\;\int \int \;g_{1}{(x}^{\prime })e^{iS(p)}e^{i(x-x^{\prime })p}dpdx^{\prime }$$
Now consider
(37)$$M_{2}\left(\theta \right)=\int {g_{2}^{*}\left(x\right)g}_{2}(x+\theta )dx$$
Straightforward substitutions of Equation (Equation 36) into Equation (Equation 37) give that
(38)$$M_{2}\left(\theta \right)=\int {g_{1}^{*}\left(x\right)g}_{1}(x+\theta )dx$$
That is
(39)$$M_{2}\left(\theta \right)=M_{1}\left(\theta \right)$$
which shows that we can generate an infinite number of g(x) in the Khintchine criteria from a given g(x) by choosing any phase function, S(p), in Equation (Equation 31).

### 4.2. Quantum Generalization of the Khintchine Criterion

Recall from Section 2 that the probability density for the random variable *a* is given by
(40)$$P\left(a\right)={\left|c(a)\right|}^{2}$$
The characteristic function is
(41)$$M_{a}\left(\theta \right)=\int {\left|c(a)\right|}^{2}e^{i\theta \alpha }d\alpha =\int c^{*}\left(a\right)c\left(a\right)e^{i\theta \alpha }d\alpha$$
We now insert a delta function
(42)$$\delta \left(a-a^{\prime }\right)=\int u_{a^{\prime }}^{*}\left(x\right)u_{a}\left(x\right)dx$$
to obtain
(43)$$M_{a}\left(\theta \right)=\;\int \int \int \;c^{*}\left(a^{\prime }\right)c\left(a\right)u_{a^{\prime }}^{*}\left(x\right)u_{a}\left(x\right)e^{i\theta \alpha }dadxda^{\prime }$$
Using
(44)$$e^{i\theta \alpha }u_{a}\left(x\right)=e^{i\theta A}u_{a}\left(x\right)$$
we have
(45)$$M_{a}\left(\theta \right)=\;\int \int \int \;c^{*}\left(a^{\prime }\right)u_{a^{\prime }}^{*}\left(x\right)e^{i\theta A}c\left(a\right)u_{a}\left(x\right)dxd\alpha^{\prime }d\alpha$$
However
(46)$$g\left(x\right)=\int c\left(a\right)u_{a}\left(x\right)da$$
and therefore
(47)$$M_{a}\left(\theta \right)=\int g^{*}\left(x\right)e^{i\theta A}g\left(x\right)dx$$

We define the characteristic function operator by
(48)$$M_{a}\left(\theta \right)=e^{i\theta A}$$
in which case,
(49)$$M_{a}\left(\theta \right)=\langle e^{i\theta A}\rangle$$
Therefore, a generalization of the Khintchine criterion is that M(θ) is a characteristic function if and only if for a self adjoint operator A there exists the representation given by Equation (Equation 47).

**Proof.** We now formally prove the sufficiency and necessity for Equation (Equation 47). The probability distribution is given by
(50)$$P\left(a\right)=\frac{1}{2\pi }\int e^{-i\theta a}\left(\int g^{*}\left(x\right)e^{i\theta A}g\left(x\right)dx\right)d\theta$$
substituting Equation (Equation 46) into Equation (Equation 50) we obtain that
(51)$$P\left(a\right)={\left|c(a)\right|}^{2}$$
which proves the sufficiency. We note that ${\left|c(a)\right|}^{2}$ is normalized if ${\left|g(x)\right|}^{2}$ is normalized. Consider now the necessity. We start with Equation (Equation 47) and define the characteristic function the usual way
(52)$$M_{a}\left(\theta \right)=\frac{1}{2\pi }\int e^{-i\theta a}{\left|c(a)\right|}^{2}da$$
Substituting for c(a) as given by Equation (Equation 5) we obtain
(53)$$\begin{array}{cc} M_{a}\left(\theta \right) & =\;\int \int \int \;e^{i\theta a}\psi \left(x^{\prime }\right)u_{a}^{*}\left(x^{\prime }\right)\;\psi^{*}\left(x\right)u_{a}\left(x\right)\;dxdx^{\prime }da \end{array}$$
(54)$$\begin{array}{cc} \; & =\;\int \int \int \;\psi \left(x^{\prime }\right)u_{a}^{*}\left(x^{\prime }\right)\;\psi^{*}\left(x\right)e^{i\theta A}u_{a}\left(x\right)\;dxdx^{\prime }da \end{array}$$
(55)$$\begin{array}{cc} \; & =\;\int \int \int \;\;\psi^{*}\left(x\right)e^{i\theta A}\psi \left(x^{\prime }\right)\delta \left(x-x^{\prime }\right)dxdx^{\prime } \end{array}$$
which gives Equation (Equation 47). □

### 4.3. Expectation Values

Using Equation (Equation 12) we have
(56)$$\begin{array}{cc} \langle a^{n}\rangle & =\frac{1}{i^{n}}\frac{d^{n}}{d\theta^{n}}M(\theta )\;{|}_{\theta =0} \end{array}$$
(57)$$\begin{array}{cc} \; & =\frac{1}{i^{n}}\frac{d^{n}}{d\theta^{n}}\int g^{*}\left(x\right)e^{i\theta A}g\left(x\right)dx{|}_{\theta =0} \end{array}$$
(58)$$\begin{array}{cc} \; & =\int g^{*}\left(x\right)A^{n}g\left(x\right)dx \end{array}$$
which is the quantum mechanical way of calculating expectation values.

### 4.4. Time Dependence

If we have an operator, A(t), that is time dependent and Hermitian for all time, then the time dependent characteristic function, M(θ,t), defined by
(59)$$M\left(\theta ,t\right)=\int g^{*}\left(x,0\right)e^{i\theta A(t)}g\left(x,0\right)dx$$
is a proper characteristic function for all times. If the operator satisfies the Heisenberg’s equation of motion
(60)$$A\left(t\right)=e^{itH}A\left(0\right)e^{-itH}$$
then
(61)$$\begin{array}{cc} M(\theta ,t) & =\int g^{*}\left(x,0\right)\exp\left[i\theta e^{itH}A\left(0\right)e^{-itH}\right]g\left(x,0\right)dx \end{array}$$
(62)$$\begin{array}{cc} \; & =\int g^{*}\left(x,0\right)e^{itH}e^{i\theta A(0)}e^{-itH}g\left(x,0\right)dx \end{array}$$
(63)$$\begin{array}{cc} \; & =\int g^{*}\left(x,t\right)e^{i\theta A(0)}g\left(x,t\right)dx \end{array}$$
where, as expected,
(64)$$g\left(x,t\right)=e^{-itH}g\left(x,0\right)$$

## 5. Born Rule by Way of Characteristic Function and Discrete Case

We now consider the case where the eigenvalues of the operator are discrete (The case of spin is particularly interesting, and in regard to Wigner distributions, the characteristic function has been previously calculated [12]). Although the previous results for the continuous case can be repeated for the discrete case, we give a different perspective where we derive quantum properties just from the characteristic function,
(65)$$M\left(\theta \right)=\langle e^{i\theta A}\rangle =\int \;\psi^{*}\left(x\right)e^{i\theta A}\;\psi \left(x\right)dx$$
If the operator A has a discrete spectrum, we write
(66)$$\;Au_{n}\left(x\right)=a_{n}u_{n}\left(x\right)$$
where $a_{n}$ are the discrete eigenvalues and $u_{n}\left(x\right)$ are the corresponding eigenfunctions. Since the operator is Hermitian the $a_{n}$ are real and the eigenfunctions are complete and orthogonal
(67)$$\begin{array}{cc} \int u_{n}^{*}\left(x\right)u_{k}\left(x\right)dx & =\delta_{nk} \end{array}$$
(68)$$\begin{array}{cc} \sum_{n}u_{n}^{*}\left(x\right)u_{n}\left(x^{\prime }\right) & =\delta (x-x^{\prime }) \end{array}$$
We expand the wave function as
(69)$$\psi \left(x\right)=\sum_{n}c_{n}u_{n}\left(x\right)$$
with
(70)$$c_{n}=\int \;\psi \left(x\right)u_{n}^{*}\left(x\right)dx$$
The probability distributions is then given by
(71)$$\begin{array}{cc} P_{a}\left(a\right) & =\int M_{a}{\left(\theta \right)}^{-i\theta a}\;d\theta \end{array}$$
(72)$$\begin{array}{cc} \; & =\frac{1}{2\pi }\;\int \int \;\psi^{*}\left(x\right)e^{i\theta A}\psi \left(x\right)e^{-i\theta a}dx\;d\theta \end{array}$$

Substituting Equation (Equation 69) into Equation (Equation 72) we have
(73)$$P\left(a\right)=\frac{1}{2\pi }\;\int \int \;\sum_{n,m}c_{m}^{*}u_{m}^{*}\left(x\right)e^{i\theta A}c_{n}u_{n}\left(x\right)e^{-i\theta a}dx\;d\theta$$
Using $e^{i\theta A}u_{n}\left(x\right)=e^{i\theta a_{n}}u_{n}\left(x\right),$ Equation (Equation 73) immediately simplifies to give
(74)$$P\left(a\right)=\frac{1}{2\pi }\sum_{n}{\left|c_{n}\right|}^{2}\int e^{i\theta a_{n}-i\theta a}\;d\theta$$
or
(75)$$P\left(a\right)=\sum_{n}{\left|c_{n}\right|}^{2}\delta \left(a-a_{n}\right)$$
Therefore, the only possible values for the random variables are the eigenvalues; the corresponding probabilities are
(76)$$P\left(a_{n}\right)={\left|c_{n}\right|}^{2}={\left|\int \;\psi \left(x\right)u_{n}^{*}\left(x\right)dx\right|}^{2}$$
This is precisely the Born rule with which Born initiated the probabilistic interpretation of quantum mechanics.

## 6. Sum and Product of Two Characteristic Functions

Suppose A and B are Hermitian operators then their sum, C,
(77)C=A+B
is also Hermitian. The characteristic function is then
(78)$$\begin{array}{cc} M(\theta ) & =\int \;\psi^{*}\left(x\right)e^{i\theta C}\;\psi \left(x\right)dx \end{array}$$
(79)$$\begin{array}{cc} \; & =\int \;\psi^{*}\left(x\right)e^{i\theta (A+B)}\;\psi \left(x\right)dx \end{array}$$
The simplification of Equation (Equation 79) is generally difficult. A simple case is where A and B commute
(80)[A,B]=0
In such a case they have common eigenfunctions and we may write
(81)$$Au_{\alpha }\left(x\right)\;=\alpha \;u_{\alpha }\left(x\right)$$
(82)$$Bu_{\alpha }\left(x\right)\;=\beta \left(\alpha \right)\;u_{\alpha }\left(x\right)$$
where α and β(α) are the respective eigenvalues, and $u_{\alpha }\left(x\right)$ are the common eigenfunctions. The characteristic function is then
(83)$$\begin{array}{cc} M(\theta ) & =\int \psi^{*}\left(x\right)\;e^{i\theta A}\;e^{i\theta B}\;\psi^{*}\left(x\right)\;\;dx \end{array}$$
(84)$$\begin{array}{cc} \; & =\int c^{*}\left(a^{\prime }\right)u_{\alpha^{\prime }}^{*}\left(x\right)e^{i\theta \alpha }\;e^{i\theta \beta (\alpha )}\;c\left(a\right)u_{\alpha }\left(x\right)\;d\alpha \;d\alpha^{\prime }\;dx \end{array}$$
which evaluates to
(85)$$M\left(\theta \right)=\int e^{i\theta \alpha }\;e^{i\theta \beta (\alpha )}{\left|c(\alpha )\;\right|}^{2}d\alpha$$
The probability density for α is therefore
(86)$$\begin{array}{cc} P(\alpha )\; & =\frac{1}{2\pi }\;\int \int \;M\left(\theta ,\tau \right)e^{-i\theta \alpha }\;d\theta \;d\tau \end{array}$$
(87)$$\begin{array}{cc} \; & =\frac{1}{2\pi }\;\int \int \;e^{i\theta \alpha^{\prime }}\;e^{i\theta \beta (\alpha^{\prime })}{\left|c(\alpha^{\prime })\right|}^{2}e^{-i\theta \alpha }\;d\theta d\alpha^{\prime }\; \end{array}$$
(88)$$\begin{array}{cc} \; & =\;\int \delta \left(\alpha^{\prime }+\beta \left(\alpha^{\prime }\right)-\alpha \right){\left|c(\alpha^{\prime })\right|}^{2}\;d\alpha^{\prime } \end{array}$$

Equation (88) can be simplified further by simplifying the delta function.

### 6.1. Example: Linear Combination of x and p

Consider the operator made up of a linear combination of ***x*** and p
(89)A=αx+βp
The operator is Hermitian for real α and β. Solving the eigenvalue problem
(90)$$\left(\alpha x-i\beta \frac{d}{dx}\right)u_{\lambda }\left(x\right)=\lambda u_{\lambda }\left(x\right)$$
gives
(91)$$u_{\lambda }\left(x\right)=\frac{1}{\sqrt{2\pi \beta }}\;e^{i(\lambda x-\alpha x^{2}/2)/\beta }$$
where we have normalized to a delta function. Hence, we have the following transform pairs
(92)$$F\left(\lambda \right)\;=\;\frac{1}{\sqrt{2\pi \beta }}\int \psi \left(x\right)e^{-i(\lambda x-\alpha x^{2}/2)/\beta }dx$$
(93)$$\psi \left(x\right)\;=\;\frac{1}{\sqrt{2\pi \beta }}\int F\left(\lambda \right)e^{i(\lambda x-\alpha x^{2}/2)/\beta }d\lambda$$
For the characteristic function, we have
(94)$$\begin{array}{cc} \;M(\theta )\; & =\;\langle e^{i\theta A}\rangle \end{array}$$
(95)$$\begin{array}{cc} \; & =\;\int \psi^{*}\left(x\right)\;e^{i\theta (\alpha x+\beta p)}\psi \left(x\right)\;dx \end{array}$$
(96)$$\begin{array}{cc} \; & =\;\int \psi^{*}\left(x\right)\;e^{i\theta^{2}\alpha \beta /2}\;e^{i\alpha \theta x}\;e^{i\theta \beta p}\psi \left(x\right)\;dx \end{array}$$
(97)$$\begin{array}{cc} \; & =\;\int \psi^{*}\left(x\right)\;e^{i\theta^{2}\alpha \beta /2}\;e^{i\alpha \theta x}\psi \left(x+\theta \beta \right)\;dx \end{array}$$
giving
(98)$$M\left(\theta \right)=\;\int \psi^{*}\left(x-\frac{1}{2}\theta \beta \right)\;e^{i\theta \alpha x}\psi \left(x+\frac{1}{2}\theta \beta \right)\;dx$$
The probability density for λ is then
(99)$$\begin{array}{cc} P(\lambda )\;\; & =\frac{1}{2\pi }\int M\left(\theta \right)\;e^{-i\theta \lambda }d\theta \end{array}$$
(100)$$\begin{array}{cc} \; & =\;\frac{1}{2\pi }\;\int \int \;\psi^{*}\left(x-\frac{1}{2}\theta \beta \right)\;e^{-i\theta (\lambda -\alpha x)}\psi \left(x+\frac{1}{2}\theta \beta \right)d\theta dx \end{array}$$
which simplifies to
(101)$$P\left(\lambda \right)\;\;=\;\;{\left|\frac{1}{\sqrt{2\pi \beta }}\int \psi \left(x\right)e^{-i(\lambda x-\alpha x^{2}/2)/\beta }dx\right|}^{2}={\left|F\left(\lambda \right)\right|}^{2}$$

### 6.2. Product of Two Characteristic Functions

A standard result in probability theory is that the product of two characteristic functions $M_{1}\left(\theta \right)$ and $M_{2}\left(\theta \right)$ is also a characteristic function
(102)$$M\left(\theta \right)=M_{1}\left(\theta \right)M_{2}\left(\theta \right)$$
This result is easily proven. Consider the probability distribution corresponding to M(θ),
(103)$$P\left(x\right)=\frac{1}{2\pi }\int M_{1}\left(\theta \right)M_{2}\left(\theta \right)e^{-i\theta x}d\theta$$
Using
(104)$$\begin{array}{cc} M_{1}\left(\theta \right) & =\int {e^{i\theta x^{\prime }}P_{1}(x}^{\prime }{)\;dx}^{\prime } \end{array}$$
(105)$$\begin{array}{cc} M_{2}\left(\theta \right) & =\int {e^{i\theta x^{\prime \prime }}P_{2}(x}^{\prime \prime }{)\;dx}^{\prime \prime } \end{array}$$
and substituting into Equation (Equation 103) one obtains that
(106)$$P\left(x\right)=\;\int \int \int \;\delta \left(x^{\prime }+x^{\prime \prime }-x\right){P_{1}(x}^{\prime }{)\;P_{2}(x}^{\prime \prime }{)\;dx}^{\prime }{dx}^{\prime \prime }d\theta$$
and therefore
(107)$$P\left(x\right)=\int {P_{1}(x}^{\prime }{)P_{2}(x-x}^{\prime }{)dx}^{\prime }$$
That is, for the product of two characteristic functions the corresponding probability density is the convolution of the two the probability densities. For the quantum case we have that
(108)$$P\left(a\right)=\int {\left|c_{1}\left(a^{\prime }\right)\right|}^{2}{\left|c_{2}\left(a-a^{\prime }\right)\right|}^{2}da^{\prime }$$
where
(109)$$\begin{array}{cc} c_{1}\left(a\right) & =\int \psi_{1}\left(x\right)u_{a}^{*}\left(x\right)\;dx \end{array}$$
(110)$$\begin{array}{cc} c_{2}\left(a\right) & =\int \psi_{2}\left(x\right)u_{a}^{*}\left(x\right)\;dx \end{array}$$

Consider the product of two characteristic functions as per Equation (Equation 102)
(111)$$\begin{array}{cc} \;M(\theta ) & =M_{1}\left(\theta \right)M_{2}\left(\theta \right) \end{array}$$
(112)$$\begin{array}{cc} \; & =\left(\int \psi_{1}^{*}\left(x\right)e^{i\theta A}\psi_{1}\left(x\right)dx\right)\left(\int \psi_{2}^{*}\left(x\right)e^{i\theta A}\psi_{2}\left(x\right)dx\right) \end{array}$$
Since M(θ) is a characteristic function, we should be able to write it as
(113)$$M\left(\theta \right)=\int \psi^{*}\left(x\right)e^{i\theta A}\psi \left(x\right)dx$$
for some wave function ψ(x). An interesting question (suggested by the referee) is to express ψ(x) in terms of $\psi_{1}\left(x\right)$ and $\psi_{2}\left(x\right),$ That is, we want
(114)$$\int \psi^{*}\left(x\right)e^{i\theta A}\psi \left(x\right)dx=\left(\int \psi_{1}^{*}\left(x\right)e^{i\theta A}\psi_{1}\left(x\right)dx\right)\left(\int \psi_{2}^{*}\left(x\right)e^{i\theta A}\psi_{2}\left(x\right)dx\right)$$
This does not seem to be readily tractable. As an example, consider the case of momentum
(115)$$A=\frac{1}{i}\frac{d}{dx}$$
then Equation (Equation 114) becomes
(116)$$\int \psi^{*}\left(x\right)\psi \left(x+\theta \right)dx=\left(\int \psi_{1}^{*}\left(x\right)\psi_{1}\left(x+\theta \right)dx\right)\left(\int \psi_{2}^{*}\left(\theta \right)\psi_{2}\left(x+\theta \right)dx\right)$$
For the case where $\psi_{1}\left(x\right)$ and $\psi_{2}\left(x\right)$ are Gaussian then ψ is also a Gaussian.

## 7. Bochner’s Theorem and Quantum Formulation

Bochner’s criterion is that M(θ) is a characteristic function if it is positive definite. That means that for any function φ(θ)
(117)$$\;\int \int \;M\left(\theta -\theta^{\prime }\right)\varphi \left(\theta \right)\varphi^{*}\left(\theta^{\prime }\right)d\theta d\theta^{\prime }\geq 0$$
Using
(118)$$M\left(\theta -\theta^{\prime }\right)=\int e^{i(\theta -\theta^{\prime })x}P\left(x\right)dx$$
we calculate the left hand side of Equation (Equation 117)
(119)$$\begin{array}{cc} \;\int \int \;M\left(\theta -\theta^{\prime }\right)\varphi \left(\theta \right)\varphi^{*}\left(\theta^{\prime }\right)d\theta d\theta^{\prime } & =\;\int \int \int \;e^{i(\theta -\theta^{\prime })x}P\left(x\right)\varphi \left(\theta \right)\varphi^{*}\left(\theta^{\prime }\right)d\theta d\theta^{\prime }dx \end{array}$$
(120)$$\begin{array}{cc} \; & =\;\int \int \int \;P\left(x\right)e^{i\theta x}\varphi \left(\theta \right)e^{-\theta^{\prime }x}\varphi^{*}\left(\theta^{\prime }\right)d\theta d\theta^{\prime }dx \end{array}$$
(121)$$\begin{array}{cc} \; & =\int P\left(x\right){\left|\int e^{i\theta x}\varphi \left(\theta \right)d\theta \right|}^{2}dx \end{array}$$
Thus, the positivity is clear. Suppose we take *x* to be position corresponding to a wave function $\psi_{1},$ and φ a momentum wave function corresponding to $\psi_{2}\left(x\right)$
(122)$$\begin{array}{cc} P(x) & ={\left|\psi_{1}\left(x\right)\right|}^{2} \end{array}$$
(123)$$\begin{array}{cc} \psi_{2}\left(x\right) & =\frac{1}{\sqrt{2\pi }}\int e^{i\theta p}\varphi \left(p\right)dp \end{array}$$
then
(124)$$\;\int \int \;M\left(\theta -\theta^{\prime }\right)\varphi \left(\theta \right)\varphi^{*}\left(\theta^{\prime }\right)d\theta d\theta^{\prime }=\sqrt{2\pi }\int {\left|\psi_{1}\left(x\right)\right|}^{2}{\left|\psi_{2}\left(x\right)\right|}^{2}dx$$

Consider now using the generalized characteristic function, Equation (Equation 47)
(125)$$M_{a}\left(\theta \right)=\int g^{*}\left(x\right)e^{i\theta A}g\left(x\right)dx$$
and taking
(126)$$g\left(x\right)=\int c\left(a\right)u_{a}\left(x\right)\;da$$
where c(a), is the wave function in the *a* representation,
(127)$$c\left(a\right)=\int g\left(x\right)u_{a}^{*}\left(x\right)\;da$$
Substituting Equation (Equation 126) into Equation (Equation 125) we obtain that
(128)$$\;\int \int \;M\left(\theta -\theta^{\prime }\right)\varphi \left(\theta \right)\varphi^{*}\left(\theta^{\prime }\right)d\theta d\theta^{\prime }=\int {\left|c(a)\right|}^{2}{\left|\int e^{i\theta a}\varphi \left(\theta \right)d\theta \right|}^{2}da\geq 0$$
which may be considered the quantum formulation of Bochner’s criteria.

## 8. Polya Sufficiency Criteria

The Polya criteria is a sufficient criteria for a function to be a characteristic function [3,13]. It only applies to probability densities that are one-sided. If a potential M(θ) is real and hence satisfies M(−θ)=M(θ), and is convex for θ>0, then it is the characteristic function of a one sided probability density. *A* function is convex if it satisfies
(129)$$M\left(\frac{\theta_{1}+\theta_{2}}{2}\right)\leq \frac{M\left(\theta_{1}\right)+M\left(\theta_{2}\right)}{2}$$
From the point of view of the generalized characteristic function the condition is that
(130)$$\begin{array}{cc} \int \psi^{*}\left(x\right)e^{i(\theta_{1}+\theta_{2})A/2}\psi \left(x\right)dx & \leq \frac{1}{2}\left(\int \psi^{*}\left(x\right)e^{i\theta_{1}A}\psi \left(x\right)dx+\int \psi^{*}\left(x\right)e^{i\theta_{2}A}\psi \left(x\right)dx\right) \end{array}$$
(131)$$\begin{array}{cc} \; & =\frac{1}{2}\int \psi^{*}\left(x\right)\left(e^{i\theta_{1}A}+e^{i\theta_{2}A}\right)\psi \left(x\right)dx \end{array}$$

## 9. Conclusions

We have given a quantum mechanical generalization of the standard characteristic function, and have shown that the Khintchine and Bochner criteria have a simple quantum mechanical interpretation, allowing the generalization of these criteria. Moreover, we have clarified what the g(x) functions are in the Khintchine criteria, Equation (Equation 15), and have given a method to generate an infinite number of them. More importantly, we have shown that they are wave functions as used in quantum mechanics. Of course, standard probability theory does not deal with wave functions, but with probability densities directly. On the other hand, quantum mechanics deals with wave functions and obtains probabilities through them. It would be interesting to study to what extent standard probability theory may be formulated in terms of wave functions.

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
