# Peer review of "Quantum Mechanical Approach to the Khintchine and Bochner Criteria for Characteristic Functions"

_entropy, 2023, doi:10.3390/e25071042_

Round 1

Reviewer 1 Report

The author of the manuscript  presents a clear mathematical  link between  quantum mechanics and standard probability theory.   He develops some interesting theorems showing how to use Khintchine criteria to define some wave functions.     He then check that Bochner’s theorem is verified. 

The paper is well written and clear.    I suggest the work for publication.

 I have only minor comments:

-The author should provide some comments in conclusion or introduction concerning his earlier work on the subject. Specifically this concerns the work      with M.O Scully  (Found  Phys  vol 16 ,  p295, 1986) where  joint Wigner  functions were introduced for spin-1/2.   This suggested very interesting development for probabilistic representations of quantum mechanics    in particular concerning hidden-variables ( see for example Scully, PRD,  vol 28 , p2477, 1983).   The new work   again suggests  how  classical probability   and quantum mechanics    and discussing the previous works could provide some clues for future ones.

Author Response

Thank you very much for your kind comments.  I have added a section (5) that does the discrete case and mentioned and referenced the suggestion of the referee.

Reviewer 2 Report

In this very interesting brief article, intriguing connections between classical probability theory and the quantum formalism are discussed by an expert of the field. The main aim is to argue that the classical Khintchine and Bochner criteria for characteristic functions admit a suitable interpretation (and generalization) by adopting a quantum point of view.

The paper is stimulating, well-written, readily understandable and enjoyable by a broad readership (being written in a fresh, mathematically informal style).

I recommend the paper under consideration for publication in its present form.

Author Response

Thank you very much for your comments

Reviewer 3 Report

This paper presents an original and attractive view on the characteristic functions (CFs) method.

It rises many basic questions that may be addressed in the framework of the current paper:

(1) Consider a product of two CFs $M(\theta)=M_1(\theta)M_2(\theta)$.  How express the wave function $g(x)$ corresponding to $M$ in terms of $g_1(x), g_2(x)$ corresponding to $M_1, M_2$? 

(2) How to understand Polya's theorem that convexity of $M(\theta)$ is a sufficient condition to be a CF?

(3) Discuss an analog of Levy-Cramer theorem: if the wave function $g$ corresponds to Gaussian distribution than both $g_1,g_1$ correspond to Guassian distribution too, etc.

Also note a typo in (62): $i$ is missed in $e^{-i\theta' x}$.

A quantum interpretation of Polya theorem would be appreciated.

How to understand the statement that a characteristic function is convex?

Note a typo on P.4, L5 up: tehn

Author Response

I very much appreciate the very interesting points you raised. I believe I have taken all of them into account.  In particular:

  1. I have added a section regarding the Polya theorem (section 8)

2     As to the product suggestion, I have added section 6 --- but I was not able to solve the general problem.

3. As to the convex issue, I believe Eq. 131 formulates the issue

Thank you again for your very interesting points -- I continue to think about them